# Elastographic Evaluation of Thyroid Nodules in Children and Adolescents with Hashimoto’s Thyroiditis and Nodular Goiter with Reference to Cytological and/or Histopathological Diagnosis

**DOI:** 10.3390/jcm11216339

**Published:** 2022-10-27

**Authors:** Hanna Borysewicz-Sańczyk, Beata Sawicka, Filip Bossowski, Janusz Dzięcioł, Artur Bossowski

**Affiliations:** 1Department of Paediatrics, Endocrinology, Diabetology with Cardiology Unit, Medical University of Bialystok, 15-089 Białystok, Poland; 2Student Research Group, Department of Paediatrics, Endocrinology, Diabetology with Cardiology Unit, Medical University of Bialystok, 15-089 Białystok, Poland; 3Department of Human Anatomy, Medical University of Bialystok, 15-089 Białystok, Poland

**Keywords:** children, thyroid nodules, thyroid cancer, nodular goiter, autoimmune thyroiditis, Hashimoto’s thyroiditis, elastography

## Abstract

There are data indicating the coexistence of papillary thyroid carcinoma and autoimmune thyroiditis (AIT) in children. The aim of the study was elastographic evaluation of thyroid nodules in children and adolescents with AIT and nodular goiter in relation to cytological and/or histopathological diagnosis. We examined 215 children (57 boys and 158 girls) with 261 thyroid nodules (143 non-AIT and 118 AIT). All study participants underwent a conventional ultrasound examination with elastography followed by fine needle aspiration biopsy (FNAB). Abnormal Strain Ratio (SR ≥ 5) was observed in 36 non-AIT nodules and 15 AIT nodules. Papillary thyroid carcinoma was diagnosed in 5 patients (2% of all investigated nodules). SR of malignant thyroid nodules was statistically higher in comparison to SR of benign nodules both in the group of non-AIT (6 ± 4 vs. 3.67 ± 2.62, *p* = 0.024) and AIT nodules (6.3 ± 0.01 vs. 2.92 ± 1.89, *p* = 0.047). Comparison of non-AIT and AIT benign nodules revealed that SR was higher in non-AIT nodules (3.67 ± 2.62 vs. 2.92 ± 1.89, *p* = 0.01). We observed a strong positive correlation (R = 1) between TSH concentration and SR ratio in the group of all malignant thyroid nodules. Autoimmune inflammatory process of the thyroid gland does not limit the use of elastography in the diagnosis of thyroid nodules in children.

## 1. Introduction

Chronic autoimmune thyroiditis (AIT) is the leading cause of thyroid dysfunction in children [1]. In recent years, an increase in the prevalence of Hashimoto’s thyroiditis (HT), which is the most common form of thyroiditis in paediatric patients, has been observed [1,2,3]. AIT is an organ-specific, autoimmune disease, characterised by autoimmune-mediated destruction of the thyroid gland [4]. The diagnosis is suggested by a characteristic ultrasound pattern and/or presence of antithyroid antibodies [1,4] or cytology results. At diagnosis, patients may be euthyroid, may have subclinical hypothyroidism, hypothyroidism or, in rare cases, initial transient hyperthyroidism [1]. Along with considerable alterations in thyroid function, AIT in children may be accompanied by structural changes in the thyroid gland [5] as the disease involves gradual atrophy of follicular tissue secondary to infiltration of lymphocytic cells and fibrotic changes [6]. Moreover, some patients develop thyroid nodules in the course of autoimmune thyroiditis [6], which might potentially be associated with an increased risk of malignancy.

Research indicates that thyroid nodules in children carry a greater risk of malignancy compared to adults [5]. Although thyroid cancer (TC), with papillary thyroid cancer (PTC) being the most common histological type, is a rare childhood malignancy with the reported incidence of around 0.59 cases per 100,000 each year, its incidence is increasing at a rate of approximately 1% per year [7]. The coexistence of TC and AIT has been observed [8], and the autoimmune process has been indicated as a risk factor for the development of TC [1,6,8]. This finding may influence the management and follow-up of patients with this disease. Thus, it is recommended that every child with HT should have a thyroid ultrasound examination at least once a year [9], and any nodules identified as suspicious by ultrasonography (US) will require a fine-needle aspiration biopsy (FNAB) to exclude malignancy.

Thyroid US is the most commonly used diagnostic method for visualizing the macroscopic structure of the thyroid gland. The sonographic appearance of the thyroid gland varies depending on the phase and severity of AIT. Typical sonographic features of HT include diffuse heterogeneity, hypoechogenic background, hypoechogenic areas, hyperechogenic septations, and hypervascularity [6]. Furthermore, US allows for the detection and characterisation of very small, impalpable thyroid nodules. The ultrasound features of thyroid nodules suggesting thyroid cancer are: hypoechogenicity, irregular shape without a halo, a taller-than-wide shape, presence of microcalcifications, intranodular vascularity greater than peripheral vascularity, and cervical lymph node enlargement [10,11,12,13,14]. The presence of a few suspicious sonographic features increases the risk of malignancy, although no characteristic appears sufficiently sensitive or specific in isolation to identify all malignant nodules [15].

Ultrasound elastography, usually performed as an extension of conventional US, is gaining interest in ultrasonographic diagnosis of thyroid lesions since it provides new details about the examined tissue and appears to increase the accuracy of US in differential diagnosis of thyroid nodules [16,17]. The mechanical properties of tissue, such as tissue stiffness, are features that reflect the nature of a thyroid lesion. In the gland that is inflamed or in which cancer has formed, tissue composition and structure are changed, which influences the parenchymal stiffness [16,18]. As demonstrated by a number of authors, most malignant thyroid nodules (e.g., papillary carcinoma) are hard, with low elasticity on palpation due to the presence of increased amounts of collagen and increased numbers of myofibroblasts [19,20]. Studies show that in the course of HT, thyroid parenchymal stiffness might be increased due to lymphocytic infiltration and variable degrees of fibrosis of the thyroid, which may affect the cohesiveness of the gland [21]. Two main thyroid elastography methods are used in clinical practice: strain elastography (SE) and shear-wave elastography (SWE). The former assesses tissue elasticity through tissue displacement induced by compression, and the latter, also called dynamic elastography, assesses tissue elasticity by measuring the propagation speed of transverse shear waves [22]. Thyroid elastography has recently been recommended as a useful complementary tool to the conventional US to improve specificity and to monitor lesions previously diagnosed as benign at FNAB [16,23]. As elastography provides information about which part of the gland has decreased elasticity, it allows for determination of the area of thyroid tissue or nodule that may require further, invasive diagnostic evaluation [16]. Furthermore, some studies indicate that elastography may help identify benign nodules and thus reduce the number of unnecessary thyroid biopsies [23,24,25].

Since the coexistence of thyroid nodules and HT, and the risk of papillary carcinoma in children with HT has been reported, the aim of our study was to determine the elastographic features of thyroid nodules in children and adolescents with HT in relation to cytological and/or histopathological diagnosis, and to assess if the autoimmune process reduces the accuracy of elastography in thyroid nodule evaluation.

## 2. Materials and Methods

### 2.1. Patients

A total of 261 thyroid nodules (118 AIT and 143 non-AIT) from 215 children and adolescents 5 to 18 years of age were included in the study conducted at the Department of Paediatrics, Endocrinology, Diabetology with Cardiology Unit, Medical University of Bialystok, Poland. Among patients with AIT nodules, only patients with HT were included in the study. The diagnosis of AIT was based on a typical ultrasound pattern and/or presence of antithyroid antibodies or cytology results. The study was granted ethical approval by the Ethics Committee of the Medical University of Bialystok, Poland. Written informed consent was obtained from the parents of participants under the age of 16, and from participants over the age of 16. The exclusion criteria were: clinical or laboratory diagnosis of Graves’ disease, a cyst detected by ultrasonography, and lack of consent to participate in the study. Table 1 shows characteristics of the study group.

### 2.2. Serum Analysis

Blood samples were collected for analysis from the basilic vein in the morning following an overnight fast and centrifuged for 10 min at 2000 rpm. Serum levels of thyrotropin (TSH) and free thyroxine (fT4) were determined by electrochemiluminescence “ECLIA” with a Cobas e 411 analyzer (Roche Diagnostics, Warsaw, Poland). Normal values for TSH ranged between 0.28 and 5.0 μIU/mL, and between 1.1 and 1.7 ng/dl for fT4. Thyroperoxidase (TPO) and thyroglobulin (Tg) antibodies titers were measured using electrochemiluminescence “ECLIA” with Modular Analytics E170 analyzer (Roche Diagnostics). Negative values for TPO antibodies were less than 34 IU/mL and negative values for Tg antibodies were less than 115 IU/mL.

### 2.3. Thyroid Imaging—Ultrasonography and Elastography

Conventional ultrasonography and elastography were performed in all patients using Toshiba Aplio MX SSA-780A system equipped with a 12 MHz linear transducer by the same, experienced ultrasonographer. Initially, the echotexture of the thyroid gland and nodules were evaluated using B-mode and Doppler ultrasound imaging. Following the assessment of sonographic characteristics, elasticity of the nodules was assessed using strain elastography. Elastography was performed by real-time, freehand method. The thyroid gland was lightly compressed and decompressed above the nodule. The process was repeated 5 times. The result was presented as the Strain Ratio (SR) which indicates deformation of the nodule, i.e., region of interest 1 (ROI 1) in comparison to the region of interest of healthy tissue (ROI 2) as a reference (ROI1/ROI2 index). Following the guidelines for studies in adults, SR 5 was taken as the cut-off value for thyroid elastography in our paediatric patients, where SR 5 or more indicated an abnormal elastography result and an increased risk of malignancy. The results were also presented in the form of a colour maps, called elastograms, which showed soft tissue in red and hard tissue in blue. Importantly, cystic elements and calcifications were excluded from the elastographic evaluation since they may have falsified the test result.

### 2.4. Fine Needle Aspiration Biopsy

An ultrasound-guided FNAB was performed in each patient with a 23-Gauge needle attached to a 5-mL disposable plastic syringe. Aspirates were smeared onto glass slides and immediately fixed in 95% alcohol for both H&E staining and May-Grunwald-Giemsa staining. A smear was considered adequate when 6 clusters of cells with >10 cells per cluster were present. Cytological diagnosis was made by an experienced pathologist and presented according to the Bethesda system. If clinically indicated (thyroid nodules within Bethesda categories VI, V, IV, or III accompanied by abnormal ultrasound findings), the patient was treated surgically and the final diagnosis was made based on histopathological results.

### 2.5. Data Analysis

Biostatistical analysis of collected data was performed using GraphPad Prism 9.0 statistical software (GraphPad Prism Inc., San Diego, CA, USA). Due to the non-Gaussian distribution of data, the nonparametric Mann-Whitney *U* test was applied to compare differences between the groups. Data were presented as mean values with SD range. Significance level was set at *p* ≤ 0.05 and differences were indicated with asterisks or exact *p* values: * −*p* < 0.05, ** −*p* < 0.01, *** −*p* < 0.001, **** −*p* < 0.0001. Correlations within the studied groups were assessed using the non-parametric Spearman’s correlation, and the obtained results were presented on heat maps as correlation coefficients (*r* values), and asterisks indicating statistically significant correlations. Depending on the value of *r*, the obtained correlations were considered weak (0.2–0.39), moderate (0.4–0.59), strong (0.6–0.79), or very strong (0.8–1.0).

## 3. Results

### 3.1. Cytological and Histopathological Results

A total of 118 single nodules (45%) out of the 261 examined nodules were AIT nodules. There were no statistically significant differences between the diagnosed AIT and non-AIT nodules in terms of size and fT4 concentration. Non-AIT patients were marginally older. TSH concentration, TPO, and Tg antibody titers were significantly higher in AIT patients (Table 1). Abnormal FNAB results—Bethesda categories III, IV, V, and VI—were obtained for 23 nodules (8 AIT and 15 non-AIT nodules), which constituted 9% of all investigated nodules. In 11 cases, the histopathological results were available following surgical treatment (ultrasound characteristic is presented in Table 2). Ultrasound surveillance was recommended for the remaining 12 nodules classified as Bethesda category III. A diagnosis of cancer was confirmed on histopathology in five patients (2% of all investigated nodules) whose nodules were classified as Bethesda categories V or VI. Two of them were AIT nodules (which constituted 1.7% of AIT nodules) and three of them were non-AIT nodules (which constituted 2.1% of non-AIT nodules). The patients diagnosed with cancer were pubertal and remained euthyroid at the time of diagnosis. One nodule classified as Bethesda category VI proved to be hyalinizing trabecular tumour/adenoma in histopathological examination, a relatively rare benign tumour whose cytological features resemble those of papillary carcinoma. One nodule classified as Bethesda category III proved to be adenoma in histopathological examination. In all patients with malignant disease, papillary thyroid carcinoma was confirmed by histopathology. Of the four patients with cancer, three were located in the right lobe, and in one was in the left lobe. Among benign, non-AIT thyroid nodules (128 nodules), the most common cytological diagnoses were: nodular goiter (40%), nodular colloid goiter (34%), nodular partially cystic goiter (11%), nodular goiter with a proliferative nodule (6%), nodular goiter with oxyphilic metaplasia (3%), a lymph node (2%), cyst (2%), and parenchymatous goiter (1%).

### 3.2. Non-AIT Thyroid Nodules

In the group of patients without AIT, malignant thyroid nodules were larger in comparison to benign nodules and the difference in size was close to statistical significance. There were no differences between benign and malignant nodules in this group in terms of laboratory results. Malignant thyroid nodules fell within a significantly higher Bethesda category (Table 3). Mean SR in patients in whom cancer was diagnosed was statistically higher than in patients with benign lesions (Figure 1). Figure 1 and Figure 2 show the ultrasound and elastography images of non-AIT nodules.

### 3.3. AIT Thyroid Nodules

Similarly to non-AIT nodules, in the group of AIT patients, malignant nodules fell within a higher Bethesda category and were bigger (significant difference). Laboratory results were similar in both groups, although malignant nodules were characterised by higher Tg titers (Table 4). The mean SR of malignant nodules was higher in comparison to benign nodules (Figure 1). Figure 3 and Figure 4 show the ultrasound and elastography images of AIT nodules.

### 3.4. Benign Thyroid Nodules

Non-AIT lesions were bigger in comparison to benign AIT nodules. Bethesda categories were similar in both groups. TSH concentrations and antithyroid antibody titers were higher in benign AIT nodules while fT4 concentrations were comparable in both groups (Table 5). SR was higher in non-AIT nodules (Figure 2).

### 3.5. Malignant Thyroid Nodules

There were no significant differences in size, Bethesda classification, laboratory results, and SR between malignant AIT and non-AIT nodules (Table 6, Figure 3).

In addition, we observed a strong positive correlation (R = 1) between TSH concentration and SR ratio in the group of all malignant thyroid nodules (Figure 4).

### 3.6. Accuracy of Elastography

Elastography revealed abnormal SR (5 and above) in 36 non-AIT nodules and 15 AIT nodules. The sensitivity of the method in non-AIT nodules was 67% and in AIT nodules 100%. The specificity of elastography in detecting malignant nodules in non-AIT patients was 76% and in AIT patients 88%. In non-AIT nodules, NPV was 99% and in AIT nodules, NPV was 100%.

## 4. Discussion

The relationship between AIT and TC has been the focus of considerable research aimed at determining the potential impact of thyroid autoimmunity on the clinical outcome of TC [26,27]. It is indicated that chronic inflammation in HT patients may provide a mutagenic environment through the contribution of intratumoural T and B lymphocytes to the evolution of TC [26,27]. According to various authors, the prevalence of thyroid nodules in children with AIT ranges from 12.2% to 31.5% [1,28,29,30,31,32]. However, data regarding the relationship between AIT and risk of TC development in children are inconsistent. It has been reported that, in adults, the coexistence of thyroid nodules and AIT increases the incidence of TC [33,34,35,36]. Moreover, it has been demonstrated that cancer in adult patients with AIT is diagnosed at a more advanced stage and that the tumour is more often multicentric [4,33,37]. With regard to paediatric patients, the correlation between AIT and TC is still under discussion. Based on a review of reports published in the years 2000–2020, Sur et al. concluded that the development of PTC in children with HT appeared to be higher than in the healthy population [38]. In Poland, a tenfold increase in the coexistence of AIT and TC was observed in children in the years 2001–2015 in comparison to the 15-year period before the year 2000 [39]. By contrast, Radetti et al. found that HT contributes to the development of thyroid nodules, but not cancer in children and adolescents [29]. Similar observations have been reported by Ben-Skowronek et al. [40]. In the present study, we observed 1.7% incidence of TC in patients with thyroid autoimmunity and 2.1% incidence in patients without autoimmune thyroid disease. According to other authors, the rate of TC among AIT children ranges from 0.67% to 9.6% [1,28,30,31,32]. In children with thyroid nodules, the percentage of TC is estimated at 22% [41], while in children with HT and thyroid nodules it ranges from 5.13 to 35% [38]. Therefore, because of the potential link between AIT with TC, careful follow-up of HT patients is necessary and polish recommendations state that children with HT should have thyroid US screening at least once a year [9].

The importance of ultrasound in guiding decisions regarding thyroid biopsies is emphasised by a number of authors. Studies in adults have shown that the use of elastography, as well as elastograhy combined with the Thyroid Image Reporting And Data System (TIRADS) classification, significantly increases the sensitivity and negative predictive value of the diagnosis of indeterminate thyroid nodules [42,43]. In our previous work, we demonstrated that elastography, as an additional diagnostic method, may improve the accuracy of the US differential diagnosis of thyroid nodules in children [44]. However, it is still a matter of debate whether the coexisting thyroid autoimmune inflammation might influence the results of the elastographic evaluation of thyroid nodules. There are studies indicating that the elasticity of autoimmune thyroid tissue is amended. Menzilcioglu et al. found significantly higher SR in HT adult patients in comparison to normal thyroid parenchyma in real-time ultrasound elastography [45]. Similar observations have been made in reference to the use of quantitative elastographic analysis evaluated by shear wave elastography (SWE) in adults [23,46]. Studies in paediatric patients have revealed that SWE scores in AIT children are significantly higher compared to paediatric patients with healthy thyroid tissue [47,48].

There are few studies in the literature evaluating the use of elastography in the differential diagnosis of thyroid nodules in AIT patients. In our study, we detected a higher SR of malignant thyroid nodules in comparison to benign nodules, both in non-AIT and AIT nodules. Moreover, the SR ratio revealed a strong positive correlation with TSH concentration, a high concentration of which is known for its carcinogenic effect. Similar observations were made by Magri et al. who reported a higher SR of malignant nodules compared to benign nodules in adult patients with and without autoimmune disease [49]. Liu BJ et al. also observed significant differences between benign and malignant nodules in HT adult patients in elastography [50].

When non-AIT and AIT benign nodules were compared in our investigation, SR was higher in non-AIT nodules, which may be due to the lower elasticity of the surrounding thyroid tissue in autoimmune disease. However, a different study by Magri et al. demonstrated similar elasticity of benign thyroid nodules in SWE in HT and non-HT adult patients, although in HT, the stiffness of extra-nodular tissue increased in relation to the thyroid antibody titre [51].

The present study demonstrated high sensitivity (100%) and specificity (88%) of elastography in detecting malignancy in AIT nodules. The values in the non-AIT group were comparable. Moreover, the method proved to have high NPV in both analysed groups (99% in non-AIT nodules and 100% in AIT nodules), denoting that a low SR in elastography in children indicates nodules that do not require invasive diagnostic procedures.

A limitation in using strain elastography for the diagnosis of thyroid nodules in autoimmune thyroiditis may be difficulty in finding healthy tissue (ROI 2) within the thyroid gland to which the elasticity of the nodule is compared (SR) since inflammation is commonly found in the entire thyroid gland. Therefore, studies involving shear wave elastography, a user-independent ultrasound technique that does not require comparison to healthy tissue, will not be burdened with this limitation.

Another consideration is the SR cut-off value, which has not been established in the paediatric population. Following the guidelines for studies in adults, we took the value of SR 5 as a cut-off point for thyroid elastography in our paediatric patients. However, further studies are needed to determine the optimal SR cut-off value for children [52,53].

A major limitation of the present study is a small sample size which is due to the low prevalence of TC in children. Since postoperative histopathological diagnosis was available only in 11 cases out of the 261 examined nodules, the prevalence of TC may have been underestimated.

## 5. Conclusions

In conclusion, elastography is characterised by high sensitivity, specificity, and NPV both in AIT and non-AIT children, and the autoimmune inflammation process in the course of HT does not decrease the accuracy of elastography in differential diagnosis of thyroid nodules. Thus, performance of elastography along with regular ultrasound examinations in children with AIT might improve diagnosis of thyroid nodules.

## Data Availability

Data supporting the reported results are available on request in University Children Hospital in Bialystok, Poland.

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
