# Peer review of "Elastographic Evaluation of Thyroid Nodules in Children and Adolescents with Hashimoto’s Thyroiditis and Nodular Goiter with Reference to Cytological and/or Histopathological Diagnosis"

_jcm, 2022, doi:10.3390/jcm11216339_

Round 1
Reviewer 1 Report
The paper titled “Elastographic evaluation of thyroid nodules in children and adolescents with Hashimoto's thyroiditis and nodular goiter with 3 reference to cytological and/or hisptopathological diagnosis” from Borysewicz-Sańczyk H. and coworkers evaluates the role of elastography in the assessment of thyroid nodules in childhood and adolescence. The topic is intriguing because it is largely discussed in thyroid literature with few data referring to pediatric population. So the paper could be of scientific interest but it is not always well written, requiring an English speaker revision. Moreover, the following comments should be addressed:
Major comments
1. The Discussion lacks a comparison of TC incidence in pediatric age with and without HT and a speculation of the underlying pathophysiological mechanisms that potentially explain the higher incidence found in HT patients; add a paragraph in the discussion section;
2. The Discussion should be implemented describing what is already known in literature about the role of elastography in thyroid nodules assessment and prediction of malignancy in adult population (e.g. Celletti I. et al La Radiologia Medica 2021);
3. The major limitation of the study is the included population: among 261 nodules examined, histopathological diagnosis was available only for 11 cases after surgical treatment; so, the prevalence of TC could have been underestimated. Moreover, the analysis has been performed in a very limited number of patients. This must be stated as an important limitation of the study and a comment should be added;
4. US features of the nodules are not described. Since it is very well known which ones are suggestive of malignancy, it would be useful to add US details. Could the author provide a table summarizing these details for the study population?
Minor comments
1. Introduction, line 48: “The comorbidity of TC with AIT has been observed”, the author should provide references for this sentence;
2. Very recently, a meta-analysis has been performed to address which thyroid nodule US features have the highest accuracy in predicting malignancy in the transition age (Cozzolino A et al. Eur Thyroid J 2022). It should be added in the reference list.
Author Response
Responses to Reviewers’ Comments
Manuscript ID: jcm-1937364
Type of manuscript: Article
Title: Elastographic evaluation of thyroid nodules in children and adolescents with Hashimoto's thyroiditis and nodular goiter with reference to cytological and/or hisptopathological diagnosis.
Authors: Hanna Borysewicz-Sańczyk, Beata Sawicka, Filip Bossowski, Janusz Dzięcioł, Artur Bossowski
Submitted to section: Endocrinology & Metabolism
Special Issue: Hypothyroidism: Causes, Effects and Current Treatments
Major comments
- The Discussion lacks a comparison of TC incidence in pediatric age with and without HT and a speculation of the underlying pathophysiological mechanisms that potentially explain the higher incidence found in HT patients; add a paragraph in the discussion section;
Thank you for that suggestion. The paragraph was added to the discussion section.
- The Discussion should be implemented describing what is already known in literature about the role of elastography in thyroid nodules assessment and prediction of malignancy in adult population (e.g. Celletti I. et al La Radiologia Medica 2021);
Thank you for that remark. This is a valuable suggestion. The discussion was supplemented with information on the use of elastography in adults.
- The major limitation of the study is the included population: among 261 nodules examined, histopathological diagnosis was available only for 11 cases after surgical treatment; so, the prevalence of TC could have been underestimated. Moreover, the analysis has been performed in a very limited number of patients. This must be stated as an important limitation of the study and a comment should be added;
Thank you for that remark. Indeed, the size of the study group is small, thus we added this information as a limitation to the manuscript.
- US features of the nodules are not described. Since it is very well known which ones are suggestive of malignancy, it would be useful to add US details. Could the author provide a table summarizing these details for the study population?
Thank you for that suggestion. The table have been added.
Minor comments
- Introduction, line 48: “The comorbidity of TC with AIT has been observed”, the author should provide references for this sentence;
Thank you for that remark. The reference have been added.
- Very recently, a meta-analysis has been performed to address which thyroid nodule US features have the highest accuracy in predicting malignancy in the transition age (Cozzolino A et al. Eur Thyroid J 2022). It should be added in the reference list.
Thank you for that suggestion. The reference have been added.

Reviewer 2 Report
Borysewicz-Sanczyk et al have done a retrospective study about the role of elastosonography in children and adolescents with Hashimoto’s thyroiditis and nodular goiter.
The article is well-written and interestingly discusses the utility of elastography in the differential diagnosis of thyroid nodules in AIT patients. There is only some lack in the methodology and in the description of the results that should be improved.
I have only minor suggestions:
1) Table 1:
-ATG: I suggested to replace with Tg. Please change accordingly in the text and in the tables.
-Size: Size I and Size II is not fully explained. What does it mean? Could you please characterize better adding a number. I suggest for example size I, referring to < 5 mm and Size II, referring to >5 mm.
2) Results: See Page 4 line 151.The choice of the title should be concordant with the description of the results. Please add an initial sub-title after the first description of your analysis.
-Figure legend 1a: See page 5. Please add in the description: Representative example of
-Table 3: Please change dimension I and II with size I and II.
-Graph 3: Please better define the statistical analysis did you used and by which methodology.
3) Discussion.
-Pag 9 line 235: Please replace course with outcome.
The authors must add some references. The human debate between thyroiditis and thyroid cancer remains controversial but a new recently established mouse model shed light on this matter.
Page 9 line 236: Please add accordingly the following references: 1) Pani F et al 2021 Endocrinology DOI : 10.1210/endocr/bqab144 ; 2) Pani F et al 2022 Cancers DOI : 10.3390/cancers14174287.
-Pag 9 line 251: Please replace immunization with autoimmunity.
Author Response
Responses to Reviewers’ Comments
Manuscript ID: jcm-1937364
Type of manuscript: Article
Title: Elastographic evaluation of thyroid nodules in children and adolescents with Hashimoto's thyroiditis and nodular goiter with reference to cytological and/or hisptopathological diagnosis.
Authors: Hanna Borysewicz-Sańczyk, Beata Sawicka, Filip Bossowski, Janusz Dzięcioł, Artur Bossowski
Submitted to section: Endocrinology & Metabolism
Special Issue: Hypothyroidism: Causes, Effects and Current Treatments
1) Table 1:
-ATG: I suggested to replace with Tg. Please change accordingly in the text and in the tables.
Thank you for that suggestion. ATG was replaced with Tg.
-Size: Size I and Size II is not fully explained. What does it mean? Could you please characterize better adding a number. I suggest for example size I, referring to < 5 mm and Size II, referring to >5 mm.
Thank you for that remark. Size I (length) and size II (width) refer to the dimensions of the thyroid nodule on US in B-mode projection. The description was clarified.
2) Results: See Page 4 line 151.The choice of the title should be concordant with the description of the results. Please add an initial sub-title after the first description of your analysis.
Thank you for that suggestion. The sub-title was added.
-Figure legend 1a: See page 5. Please add in the description: Representative example of
Thank you. The description was added.
-Table 3: Please change dimension I and II with size I and II.
Thank you for that remark. Changed.
-Graph 3: Please better define the statistical analysis did you used and by which methodology.
Thank you for that suggestion. It was clarified.
3) Discussion.
-Pag 9 line 235: Please replace course with outcome.
Thank you for that suggestion. It was changed.
The authors must add some references. The human debate between thyroiditis and thyroid cancer remains controversial but a new recently established mouse model shed light on this matter.
Page 9 line 236: Please add accordingly the following references: 1) Pani F et al 2021 Endocrinology DOI : 10.1210/endocr/bqab144 ; 2) Pani F et al 2022 Cancers DOI : 10.3390/cancers14174287.
Thank you for that remark. This is a valuable suggestion. The references were added.
-Pag 9 line 251: Please replace immunization with autoimmunity.
Agree. It was replaced.

Reviewer 3 Report
In this study Borysewicz-Sańczyk et al, analyzed the elastographic features of thyroid nodules in children and adolescents with and without an autoimmune thyroid disease using cytology histopathology as reference standard. The Authors found that SR was significantly higher in malignant nodules independently from the presence or not of an autoimmune disease.
The study could be clinically relevant. However, there are some major criticisms:
1. The English of the manuscript must be improved
2. Table 1: parameters showed in table 1 seem expressed as mean ± SD and by a range. Please indicate clearly in the table legend how data are expressed. Furthermore, please indicate what the term “size I and II” refers to.
3. The Authors indicated as strain ratio cut off the value of 5. Since this a crucial parameter, the Authors must indicate how this cut off was calculated.
4. What were the indications for surgery?
5. Data analysis: the Authors indicated that “Due to non-Gaussian distribution of the data, nonparametric Mann-Whitney U test was applied to compare differences between groups. Data were presented on the graphs as median values with inter quartile range”. However, all data included in tables and graphs seem indicated as mean ± SD.
6. Figures should be mentioned also in the text
Author Response
Responses to Reviewers’ Comments
Manuscript ID: jcm-1937364
Type of manuscript: Article
Title: Elastographic evaluation of thyroid nodules in children and adolescents with Hashimoto's thyroiditis and nodular goiter with reference to cytological and/or hisptopathological diagnosis.
Authors: Hanna Borysewicz-Sańczyk, Beata Sawicka, Filip Bossowski, Janusz Dzięcioł, Artur Bossowski
Submitted to section: Endocrinology & Metabolism
Special Issue: Hypothyroidism: Causes, Effects and Current Treatments
- The English of the manuscript must be improved
Thank you. English was checked and improved.
- Table 1: parameters showed in table 1 seem expressed as mean ± SD and by a range. Please indicate clearly in the table legend how data are expressed. Furthermore, please indicate what the term “size I and II” refers to.
Thank you for that remark. The description of the data in the table was clarified.
Size I (length) and size II (width) refer to the dimensions of the thyroid nodule on US in B-mode projection. The description was clarified.
- The Authors indicated as strain ratio cut off the value of 5. Since this a crucial parameter, the Authors must indicate how this cut off was calculated.
Thank you for that remark. This section was described.
- What were the indications for surgery?
Thank you. This is a valuable suggestion. The description was added.
- Data analysis: the Authors indicated that “Due to non-Gaussian distribution of the data, nonparametric Mann-Whitney U test was applied to compare differences between groups. Data were presented on the graphs as median values with inter quartile range”. However, all data included in tables and graphs seem indicated as mean ± SD.
Thank you. It was corrected.
- Figures should be mentioned also in the text
Thank you. Figures are mentioned in the taxt.
Round 2
Reviewer 1 Report
The paper is improved in its revised version. Anyway there are some minor comments that should be addressed:
1. Table 2 - taller-than-wide shape and the presence of suspected lymph nodes should be added to the other US features, as they have been demonstrated to predict malignancy in paediatric thyroid nodules;
2. Very recently, a meta-analysis evaluating different US-elastography techniques for thyroid nodules characterisation has been published (Cantisani V. et al. Frontiers in Oncology 2022); it should be added to the reference list;
3. Conclusions, line 412 - "in children" is repeated twice, please delete;
4. Discussion, line 399 - the sample size should be pointed up as a "major" limitation of the study.
Author Response
Responses to Reviewers’ Comments
Manuscript ID: jcm-1937364
Type of manuscript: Article
Title: Elastographic evaluation of thyroid nodules in children and adolescents with Hashimoto's thyroiditis and nodular goiter with reference to cytological and/or hisptopathological diagnosis.
Authors: Hanna Borysewicz-Sańczyk, Beata Sawicka, Filip Bossowski, Janusz Dzięcioł, Artur Bossowski
Submitted to section: Endocrinology & Metabolism
Special Issue: Hypothyroidism: Causes, Effects and Current Treatments
- Table 2 - taller-than-wide shape and the presence of suspected lymph nodes should be added to the other US features, as they have been demonstrated to predict malignancy in paediatric thyroid nodules;
Thank you for that suggestion. The information was added.
- Very recently, a meta-analysis evaluating different US-elastography techniques for thyroid nodules characterisation has been published (Cantisani V. et al. Frontiers in Oncology 2022); it should be added to the reference list;
Thank you for that suggestion. The reference was added.
- Conclusions, line 412 - "in children" is repeated twice, please delete;
Thank you. It was corrected.
- Discussion, line 399 - the sample size should be pointed up as a "major" limitation of the study.
Thank you for that suggestion. We emphasized this information.

Reviewer 3 Report
The manuscript was revised along the suggested lines. All the raised comments were addressed and discussed in the revised text.
Author Response
Responses to Reviewers’ Comments
Manuscript ID: jcm-1937364
Type of manuscript: Article
Title: Elastographic evaluation of thyroid nodules in children and adolescents with Hashimoto's thyroiditis and nodular goiter with reference to cytological and/or hisptopathological diagnosis.
Authors: Hanna Borysewicz-Sańczyk, Beata Sawicka, Filip Bossowski, Janusz Dzięcioł, Artur Bossowski
Submitted to section: Endocrinology & Metabolism
Special Issue: Hypothyroidism: Causes, Effects and Current Treatments
The manuscript was revised along the suggested lines. All the raised comments were addressed and discussed in the revised text.
Thank you.